# Foundational Aspects for Incorporating Dependencies in Copula-Based Bayesian Networks Using Structured Expert Judgments, Exemplified by the Ice Sheet–Sea Level Rise Elicitation

**DOI:** 10.3390/e26110949

**Published:** 2024-11-05

**Authors:** Dorota Kurowicka, Willy Aspinall, Roger Cooke

**Affiliations:** 1Delft Institute of Applied Mathematics, Delft University of Technology, 2628 CD Delft, The Netherlands; cooke@rff.org; 2School of Earth Sciences, University of Bristol, Bristol BS8 1RJ, UK; willy@aspinallassociates.com

**Keywords:** expert judgment, Bayesian networks, copulas

## Abstract

The work presented here marks a further advance in expert uncertainty quantification. In a recent probabilistic evaluation of ice sheet process contributions to sea level rise, tail dependence was elicited and propagated through an uncertainty analysis for the first time. The elicited correlations and tail dependencies concerned pairings of three processes: Accumulation, Discharge and Run-off, which operate on major ice sheets in the West and East Antarctic and in Greenland. The elicitation enumerated dependencies between these processes under selected global temperature change scenarios over different future time horizons. These expert judgments allowed us to populate a Paired Copula Bayesian network model to obtain the estimated contributions of these ice sheets for future sea level rise. Including positive central tendency dependence and tail dependence increases the fatness of the upper tails of projected sea level rise distributions, an amplification important for designing and evaluating possible mitigation strategies. Detailing and jointly computing distributional dependencies and tail dependencies can be crucial components of good practice for assessing the influence of uncertainties on extreme values when modelling stochastic multifactorial processes.

## 1. Introduction

Many societally important issues entail substantial scientific uncertainty and their characterisation and numerical treatment for hazard or risk assessment can be challenging. It can also be fraught and compromised by limiting assumptions. Some examples of this are as follows: emergent or species-jumping zoonoses, or other nascent public health threats; industrial accidents; natural hazards, such as floods, earthquakes or volcanic eruptions; and climate change. Here, we use a special case from the last of these to illustrate how uncertainty distribution correlations and tail dependencies can influence, and avoid, mistakenly under-estimating extreme value likelihoods.

Future sea level rise (SLR) poses serious threats to the viability of many coastal communities and important facilities but is challenging to forecast over decadal to century timescales using deterministic modelling approaches. In the period following the 2013 Fifth Assessment Report (AR5) of the Intergovernmental Panel on Climate Change (IPCC) [1] and the subsequent Sixth Assessment Report (AR6) in 2023 [2], considerable advances were made in understanding ice sheet melting processes, in their numerical modelling and in the observational record of ice sheet contributions to global mean SLR. In addition, severe limitations concerning the predictive capability of ice sheet models were better recognised (e.g., [1]).

However, at the time of AR5, quantitative estimates of future melting rates of the three major global ice sheets (Greenland, West Antarctica, East Antarctica) represented the largest source of uncertainty in estimating future SLR driven by climate change. Therefore, to serve the pressing needs of the science and policy communities, it was necessary to exploit alternative approaches for assessing future SLR and, critically, the associated uncertainties.

Therefore, in the interval between the two IPCC Assessment Reports (AR5 and AR6), a structured expert judgment study was undertaken, focused on potential ice sheet contributions to SLR. New techniques were used for modelling correlations between inter- and intra-ice sheet processes and, more crucially, for quantifying tail dependencies. The relevant expert elicitation was undertaken in 2018 via separate, two-day workshops, one held in the USA and one in the UK, accessing the professional judgments of twenty-two glaciologists in total. Details of how these experts were selected are provided in [3] Supplementary Information (SI) Note S1. The protocols of the workshops and the elicitation questions were identical, so that the two sets of findings could be combined using an impartial weighting approach (see Section 2).

Structured expert judgment (SEJ)—as opposed to other types of expert elicitation or basic opinion polling—weights each expert using objective estimates of their ‘statistical accuracy’ and ‘informativeness’, determined using the experts’ uncertainty evaluations over a set of seed questions (sometimes referred to as calibration items); the questions, drawn from glaciology, were all ones with ascertainable values (see Section 2). The approach is analogous to weighting alternative climate-related models based on their ability to simulate an observed property of the climate system according as their different capabilities in capturing a relevant property, such as the regional 20th century surface air temperature record (for the specifics of the elicitation seed items see [3] SI Note 8 and doi for the available files).

In a typical SEJ, the weighted combination ‘Decision Maker’ (DM)—i.e., the performance weighted (PW) combination of all the experts’ judgments—in general outperforms an equal weights (EW) combination in terms of statistical accuracy and informativeness jointly. The approach is particularly effective at identifying those experts who can quantify their uncertainties with high statistical accuracy for specified problems rather than, for example, experts with restricted domains of knowledge or, even, those who have high scientific reputations but a questionable judgment of scientific uncertainty [4].

In the motivating study [3], participating glaciologists quantified their uncertainties for the three main physical processes influencing ice sheet mass balance: Accumulation, Discharge and surface Run-off. The experts carried this out separately for the Greenland, West Antarctic and East Antarctic ice sheets (GrIS, WAIS and EAIS, respectively) and for two schematic global surface air temperature change scenarios. The first temperature trajectory (Low) stabilises in 2100CE at +2 °C above pre-industrial global mean surface air temperature (defined as the average for 1850–1900) and the second (High) stabilises at +5 °C ([3] SI Figure S1). The experts generated process-dependent contribution values to total sea level change, with probabilistic uncertainties, for four projection dates: 2050CE, 2100CE, 2200CE and 2300CE. The experts also quantified dependencies between Accumulation, Run-off and Discharge for each ice sheet individually. But to reduce the elicitation burden on the experts, it was agreed that a consideration of process dependencies between ice sheets had to be restricted solely to Discharge, as that was regarded by the group as the most critical process in terms of contributing to sea level rise. Thus, inter-ice sheet Discharge dependencies were determined for the 2100CE High temperature rise scenario, but not for the other processes or time horizons.

Global mean surface air temperature trajectories were adopted for the elicitation, rather than CO_2_ emission scenarios. This was to isolate the experts’ judgments about the relationships between temperature change scenarios and ice sheet changes from judgments in relation to the complicating—and potentially confounding—issue of future climate sensitivity. For the elicitation, the most straightforward approach was for the experts to tie their judgments directly to scenarios related to mean global air temperature. An SEJ elicitation on CO_2_ emissions using the same methodology but without dependence elicitation is described in [5].

An important and unique element of the 2018 elicitation was the quantification of intra- and inter-ice sheet dependencies (see also [3] SI Note S1.5). Two features of dependence were elicited: a ‘central dependence’ and an ‘upper tail dependence’. The former seeks to capture the probability that one variable exceeds its median given that the other variable exceeds its median, while the latter captures the probability that one variable will exceed its 95th percentile given that the other exceeds its 95th percentile. It is well known that these two types of dependence are, in general, markedly different, a property that is not captured by the usual Gaussian dependence model [6]. The latter assumption always imposes tail independence, regardless of the degree of central dependence, and can produce large errors when applied inappropriately. For example, if—under a given temperature rise scenario—Greenland ice sheet Discharge exceeds its 95th percentile, what is the probability that Greenland Run-off will also exceed its 95th percentile? This probability may be substantially higher than the independent probability of 5%, and ignoring tail dependence, may lead to under-estimating the probability of high end SLR contributions.

Based on each expert’s responses, a joint distribution was constructed to capture the dependencies between Run-off, Accumulation and Discharge for GrIS, WAIS and EAIS with dependence structures chosen, per expert, to capture both central and tail dependencies (see [3]: Methods and SI Note S1.5).

To help interpret elicitation findings, in [3] the participating glaciologists were also asked to provide qualitative and rank-order information on what they regarded to be the leading processes for influencing ice dynamics and surface mass balance (e.g., snowfall accumulation minus ice ablation loss). For completeness, the combined sea level contribution—from all mass balance processes and from the three ice sheets—was computed, assuming either independence or dependence (further details can be found in [3] SI).

Here, focus is placed on appraising the original elicited dependencies in a more detailed way than was possible in [3] and on establishing any improvement in the uncertainty quantification of sea level rise projections, which may be achieved by incorporating more sophisticated mathematical representations of the various contributory dependencies into the Bayesian network (BN) model—relative, that is, to those in the earlier, 2019 results. The elicitation was conducted in terms of future changes to sea level due to sheet process contributions under different global temperature change scenarios. At the first workshop, it was agreed the baseline reference would be the mean 2000–2010 global annual mass transfer rate from the three ice sheets to the ocean. To express future SLR trends, experts provided their judgments in terms of sea level change relative to this baseline rate. The baseline rate is added to the aggregated SLR change judgments, integrated over the chosen future timescales. This causes the values in [3] to differ slightly from the raw elicitation values presented here. N.b., here, improvement in the quantification of SLR projections may entail an increase in quantified uncertainty, especially in distribution tails—an important potential outcome for the credibility of sea level rise estimation.

Our refined and updated analysis can be consequential because a very significant conclusion in [3] was that probability distributions with long upper tails were greatly influenced by inter-dependencies between ice sheet processes and between the three ice sheets collectively. One important result in [3] was the finding that global total SLR (including thermal expansion and glacier Run-off) exceeding 2 m by 2100CE was estimated to lie beneath the 95 percentile of the combined glaciologists’ judgments. When published in 2019, this was more than twice the upper value put forward previously for 2100CE by the Intergovernmental Panel on Climate Change in AR5.

Thus, the 95th percentile estimate in [3], with a sea level rise estimate exceeding 2 m by 2100CE, represents a substantial future societal risk. Because a sound basis for the further estimation of the probabilities of plausible upper tail rises in global sea level is justified, the imperative exists for adopting a more comprehensive appraisal and refinement of uncertainty modelling—using the most advanced Bayesian network methods and copula dependence modelling techniques.

Therefore, in the following sections, we describe and discuss the following: Section 2 methods for processing expert elicitations and for constructing related BNs; Section 3 detailed assessments of margins summarised from [3], tail dependencies and copulas and copula-based BNs; Section 4 implications of the overarching BN modelling and analysis and appraisal of the BN model performance; Section 5 a summary of revised findings pertinent to future SLR projections and Section 6 conclusions concerning the generic methodological advances intrinsic to our uncertainty modelling approach and recommendations for further work.

## 2. Materials and Methods

This section starts with the presentation of the structured expert judgment (SEJ) method, followed by basic concepts of Bayesian networks (BNs).

### 2.1. Structured Expert Judgment SEJ-Method

The experts quantified their uncertainty for the physical processes of Run-off (*R*), Accumulation (*A*) and Discharge (*D*) for the ice sheets in Greenland (*G*), West Antarctica (*W*) and East Antarctica (*E*) for each epoch (2050, 2100, 2200, 2300) and for a Low (+2 °C) and High (+5 °C) warming scenario. Expert assessments take a form of fixed percentiles, 5th, 50th, 95th, from the assessor’s subjective distribution for a each unknown quantity. In addition, experts quantified their uncertainty for 16 calibration variables from their field, for which true values were retrieved. These data were processed according to the Classical Model (CM), which constructs a performance-based weighting based on the calibration variables and compares this with equal weighting (for a recent exposition of CM concepts, see [7] or the Supplementary Information of [3]).

CM uses two measures of performance:Statistical Accuracy (SA) is measured as the probability of falsely rejecting the hypothesis that a probabilistic assessor is statistically accurate. It is, in other words, the *p*-value of rejection for this hypothesis. We hasten to add that CM does *not* test and reject expert hypotheses but, in compliance with proper scoring rule theory for sets of assessments, uses this *p*-value to measure the degree of correspondence between assessments and data in forming weighted combinations of expert distributions. When *n* true values for a number of such quantities are observed, we compute the sample distribution s of inter-quantile relative frequencies and compare this with the theoretical inter-quantile mass function p=(0.05,0.45,0.45,0.05). The test statistic is 2n times the log likelihood ratio of s and p. Assuming that the realisations are independently sampled from the assessor’s distributions, this statistic is asymptotically chi-square distributed with degrees of freedom equal to the number of assessed percentiles. Low scores (near 0) mean it is unlikely that the divergence between **s** and **p** should arise by chance. Higher scores (near 1) indicate better agreement between s and p.Informativeness (Inf) is measured as the Shannon relative information of the minimal information fit to the experts’ percentiles, relative to a user selected background measure. In this analysis, the background measure per variable is always uniform on an interval 20% larger than the smallest interval containing all experts’ assessments and the realisation. Relative information is tail insensitive and “slow” so that the experts’ information scores are quite insensitive to the size of the background measure. This slowness means that the ratio of Inf scores is much less variable over experts than the ratio of SA scores.

Dependence was considered important for the probabilistic modelling as the physical processes interact in ways not fully understood. Central dependence refers to overall dependence as measured by rank correlation; tail dependence refers to dependence in the deep tails, which is not captured by central dependence measures. These are distinct notions [6]. Variables can have a weak central dependence yet show strong association in the tails. In modelling SLR, it is especially important to capture, e.g., the probability of very high values of Run-off and Discharge as these both contribute to SLR.

### 2.2. Bayesian Networks

The qualitative part of a BN is represented by a directed acyclic graph (DAG), G=(V,E), with nodes $V=\{v_{1},\ldots ,v_{d}\}$ and directed edges (arcs) *E*. Any sequence of arcs $v_{i_{1}}\to \ldots \to v_{i_{k}}$, is called a path from node $v_{i_{1}}$ to node $v_{i_{k}}$ in *G*. The graph *G* is called *acyclic* if it does not contain a path that starts and ends at the same node. For each edge v→w in *E*, the node *v* is said to be the parent of *w* and *w* is said to be the child of *v*. Moreover, we denote as pa(v) (ch(v)) the set of parents (children) of *v*. Since *G* is a directed acyclic graph, its nodes can be ordered (order denoted as ‘<’) such that parents appear earlier in the order than the children. A simple DAG with four nodes is shown in Figure 1. In this DAG, node $v_{3}$ has two parents ($pa\left(v_{3}\right)=\left\{v_{1},v_{2}\right\}$) and one child ($ch\left(v_{3}\right)=\left\{v_{4}\right\}$). Moreover, the nodes of this DAG can be uniquely ordered $v_{1}<v_{2}<v_{3}<v_{4}$.

The nodes in *V* correspond to random variables $X_{1},\ldots ,X_{d}$. We assume that if two nodes are connected by an arc, then the corresponding variables are directly related, and if there is no arc between two nodes $v_{i}$ and $v_{j}$ and $v_{i}<v_{j}$, then corresponding variables $X_{i}$ and $X_{j}$ are conditionally independent given variables corresponding to all parents of node $v_{j}$ [8].

The quantitative part of the BN models is composed of the conditional distributions of each node, given its parents. Then, the joint distribution of $\left(X_{1},\ldots ,X_{d}\right)$ is equal to the product of these conditional distributions: (1)$$P_{V}\left(x_{V}\right)=\prod_{v\in V}P_{v|pa\left(v\right)}\left(x_{v}|x_{pa\left(v\right)}\right).$$

If all random variables $X_{1},\ldots ,X_{d}$ are discrete, then the product of conditional probability tables on the right hand side of (Equation 1) gives us the joint probability mass function of $\left(X_{1},\ldots ,X_{d}\right)$. If all $X_{i}^{\prime }$s are absolutely continuous, the conditional densities of each node given their parents are needed and $P_{V}$ represents the joint density of $\left(X_{1},\ldots ,X_{d}\right)$.

### 2.3. Gaussian BNs

The most popular applications are Gaussian BNs, where the conditional density of each node is Gaussian and the node means are a linear function of the parents and constant variances.
$$X_{v|pa(v)}∼N\left(\mu_{v}+\sum_{w\in pa(v)}b_{v}^{w}X_{w},\sigma_{v}^{2}\right).$$

Such a representation of BNs is equivalent with the specification of the joint Gaussian distribution of variables corresponding to nodes of the DAG (see e.g., [8]).

### 2.4. Copula-Based BNs

The restrictions of Gaussian BNs can be relaxed by the application of copulas. A copula is a distribution on the unit hypercube with uniform marginal distributions [6]. It contains all the information about the dependence between elements of a random vector. One can extract such a dependence structure corresponding to, e.g., multivariate Gaussian distribution in the form of a Gaussian copula and use it to construct a BN with this copula and different marginal distributions. Gaussian copula-based BNs have been successfully applied in many areas (some examples are described in [9]). Copulas other than Gaussian types of copulas are available that are able to model asymmetries and/or tail dependencies. In Figure 2, two bivariate copula densities are presented.

The Gumbel copula presented in Figure 2 is an example of copula belonging to the Archimedean family. It is asymmetric and has a property of upper tail dependence, which can be observed as a very high value of its density at the (1,1) corner of the unit square. The extensions of Archimedean copulas to higher dimensions are available but they are limited by the types of dependencies that these models can describe. In particular, the canonical Archimeaden copulas are able to only model exchangeable dependencies. In [10], Archimedean copulas were applied to represent the conditional densities in the density decomposition of BN in (Equation 1).

In [11], a very flexible copula-based BN approach was introduced. It is based on the representation of each conditional density in the factorisation in (Equation 1) using a sequence of bivariate (conditional) copulas (see also [12,13]). For a node that has more than one parent, the convention is to order its parents (construct a parental ordering) and assign several copulas. First, a copula is assigned between the node and its first parent; then, a copula is assigned between the node and its second parent, conditional on the first parent; then, a copula is assigned between the node and its third parent, conditional on the first and the second parents. This process continues for all parents of each given node. Note that all these (conditional) copulas are unconstrained, in the sense that they can be chosen independently of each other and still form a valid density.

A Paired Copula Bayesian network (PCBN) is composed of the following:(qualitative part) a DAG, *G* and a set of orders of parents of each node $O_{v}$, denoted as *O*;(quantitative part) marginal densities of variables corresponding to each node in *G* and the set of conditional copulas $c_{wv|pa(v;w)}$, where pa(v;w) denotes a set of parents of *v* before *w* in the order $O_{v}$ (for nodes without parents $c_{wv|pa(v;w)}=c_{wv}$ by convention).

The joint density $P_{V}\left(x_{V}\right)$ in (Equation 1) in case of paired copula specification, can be rewritten
(2)$$P_{V}\left(x_{V}\right)=\prod_{v\in V}P\left(x_{v}\right)\;\prod_{w\in pa\left(v\right)}c_{wv|pa(v;w)}\left(F_{w|pa(v;w)}\left(x_{w}|x_{pa\left(v;w\right)}\right),F_{v|pa(v;w)}\left(x_{v}|x_{pa\left(v;w\right)}\right)\right)$$
where the functions $F_{w|pa(v;w)}\left(x_{w}|x_{pa\left(v;w\right)}\right)$ can be computed from copulas assigned to the arcs of BN (the copulas in simplified form, i.e., not depending directly on the conditioning variables, are presented in the decomposition (Equation 2)). This distribution comprises the product of margins: i.e., the first product in (Equation 2) and the product of copulas for each node and its parents (the second product in (Equation 2)). This second product is a factorisation of the joint copula corresponding to the joint density, which can be denoted as $c_{V}$.

Since the parental set of node $v_{3}$ in the DAG in Figure 1 can be ordered as $v_{1},\;v_{2}$ or $v_{2},\;v_{1}$, then the following copulas can be assigned to the BN’s arcs:$$c_{12},c_{13},c_{23|1}\;\mathrm{and}\;c_{34}$$
or
$$c_{12},c_{23},c_{13|2}\;\mathrm{and}\;c_{34}.$$

When all the copulas in the factorisation (Equation 2) are Gaussian copulas, then the joint distribution of variables corresponding to nodes in the graph is the distribution with copula $c_{V}$, which is the joint Gaussian copula. Hence, the Gaussian copula-based BNs are a special case of PCBNs.

PCBNs require the specification of bivariate (conditional) copulas, which are unconstrained. They can belong to any parametric family or can be non-parametric.

## 3. Results SLR—Model

Below, we present a recent application of PCBN models to compute sea level rise contributions from ice sheets, following [3].

### 3.1. Assessment of Marginals

In this section, a brief summary of the revised assessments of contributions to future sea level rise due to Accumulation (A), Run-off (R) and Discharge (D) for Greenland (G), and the West (W) and East (E) Antarctic ice sheets is presented to illustrate the impact of our dependence modelling on the published findings. The results relate to the ‘High’ temperature scenario in [3], under which the global mean annual Surface Average Temperature rises by +5 °C by year 2100, with respect to pre-industrial temperatures (but, here, excluding the 2000–2010 baseline forward adjustment).

The experts provided their judgments in the form of 5th, 50th and 95th percentiles of quantities of interest (see one example below).


*In the case of Greenland, for a global temperature rise of +5 °C by 2100, what will be the integrated contribution, in mm of SLR, relative to 2000–2010 for the following:*
*(i)*   
*Accumulation*

*5% value: _______ 50% value: _______ 95% value: _______*
*(ii)*  
*Run-off*

*5% value: _______ 50% value: _______ 95% value: _______*
*(iii)* 
*Discharge*

*5% value: _______ 50% value: _______ 95% value: _______*



The experts also provided judgments on so-called seed questions, which were used to assess their performance and then to choose a weighted combination of experts to obtain distributions of SLR contributions. The full analysis of the assessments and the results can found in the SI [3]. The analysis resulted in eight experts obtaining non-zero weight. These weights are presented in Table 1.

The marginal distributions of each expert’s assessment of the contribution to sea level, due to Discharge in West Antarctica (WD) under the High (+5 °C by 2100) scenario, together with the performance-based combination, are shown in Figure 3. We can see some marked variability in expert opinions, such as is often observed in SEJ studies. More information about the expert assessments and the analysis of expert performance in this case can be found in the SI [3].

### 3.2. Assessment of Dependencies

All experts quantified the dependence between these quantities at 2100 with +5 °C global surface air temperature warming, with respect to pre-industrial temperatures. The experts were asked to assess bivariate dependencies between pairs X, Y (all combinations are presented in Figure 4 as edges of the undirected graph) in the form of the 0.5 and 0.95 exceedance probabilities (ExcProb).
(3)$$P\left(Y>q_{Y,0.5}|X>q_{X,0.5}\right),\;\;P\left(Y>q_{Y,0.95}|X>q_{X,0.95}\right)$$
where $q_{Y,0.5},q_{Y,0.95}$ denote 0.5 and 0.95 quantiles (i.e., 5th and 95th percentiles) of the distribution of *Y*, respectively.

Some experts were comfortable enough to provide assessments in the form of values of these probabilities; others preferred to use the classification presented in Table 2. All of the experts’ responses were translated to the classification scheme in Table 2 (this scheme is slightly more comprehensive than the summary Supplementary Information provided originally in Table S8 in [3]).

### 3.3. Experts’ BNs

The copulas corresponding to the assessed 0.5 and 0.95 ExcProb can be easily computed for parametric copulas with cumulative distribution function *C* as follows:$$ExcProb_{p}=\frac{1-2p+C(p,p)}{1-p}$$
where p=0.5, 0.95. The results are presented in Table 3. Frank’s copula is chosen when no tail dependence is indicated by the expert. For strong tail dependence, together with strong overall dependence, the Survival Clayton copula is preferred, and, when medium degrees of tail and of overall dependence are suggested, this situation is associated with either a Gaussian or with a Gumbel copula. In cases where an expert assessed the 0.5 ExcProb to be equal to 0.5 (as applies in case of independence) and also indicated that tail dependence is present, the t-copula is chosen.

Note that it might be not possible to choose a copula (out of five parametric families considered in this study, or even in general) that realises values of both ExcProb provided by an expert. For instance, when the 0.5 ExcProb is assessed to be 0.5 and the 0.95 ExcProb is larger than 0.05, then the t-copula, t(0,1), is chosen. In this case, the 0.5 and 0.95 ExcProb are 0.5 and 0.3, respectively. Of the parametric copulas used in this study, there is no copula that would realise the 0.5 ExcProb of 0.05 and 0.95 ExcProb as larger than 0.3.

To quantify dependencies in PCBN, the conditional copulas are computed (out of the five types of copulas discussed above), such that the 0.5 and 0.95 ExcProb for bivariate margins specified through the conditional copula are as close as possible to the probabilities specified by the expert. Assume that the experts specified $ExcProbs_{i,j,p}$, where i,j=1,2,3 and p=0.5, 0.95 for variables $X_{1},\;X_{2},\;X_{3}$. First, copulas $C_{i,3}$ with parameters $\theta_{i,3},\;i=1,2$ corresponding to $ExcProbs_{i,3,p}$ are found. Using these copulas, the conditional distributions $F_{i|3}\left(p,x,\theta_{i3}\right)$ are computed by the differentiation of $C_{i,3}$, with respect to $x_{3}$. To find a conditional copula $C_{12;3}$ with parameter $\theta_{12;3}$, such that the $ExcProbs_{1,2,p},p=0.5,0.95$ agree with ones specified by the expert, the following formula is used:$$ExcProb_{1,2,p}=\frac{1-2p+\int_{0}^{1}C_{12;3}\left(F_{1|3}\left(p,x,\theta_{13}\right),F_{2|3}\left(p,x,\theta_{23}\right),\theta_{12|3}\right)dx}{1-p}$$
where p=0.5,0.95.

The problem of the non-existence of parametric copulas to agree with experts’ assessments becomes even more pronounced when conditional copulas are specified. In the simplified form, which we are using, conditional copulas depend on the values of the conditioning variables only through the conditional cumulative distributions of the coupled variables. The form of copula does not change. A few examples of such inconsistencies are reported in Table 4 where all conditional copulas used for the eight experts’ models are presented. The inconsistent ones are printed in red. In the case of expert 27, the inconsistency stems from the fact that this expert assessed the bivariate margins (GR,GA) to be uncorrelated overall but with a strong tail dependence, yet both other margins (GD,GA) and (GR,GD) were assessed by this expert to be strong overall, as well as having tail dependence. Specifying the conditional copula GR,GD|GA to be Survival Clayton with a very high value of the parameter was not sufficient to obtain the 0.95 ExcProb of 0.7 for (GR,GD), as specified by expert 27.

In Figure 5, the PCBN models of eight experts who received non-zero weight are presented. The copulas (type and parameter value) used to model dependencies between variables are assigned to the arks of the graph. The conditional copulas are in red. We can observe that some experts included extra non-dependencies in their models: the relationship between ED and EA especially is assessed as very weak.

## 4. Sea Level Rise Estimates—Reappraisal with Dependencies

Ice sheet contributions to SLR by 2100 in [mm] for the High (+5 °C) global surface air temperature rise scenario are computed with the PCBN model. The results (see Table 5 and Appendix A) in the case where all experts’ SLR assessments are assumed to be independent and combined with equal weights (IndepSLR-EW) are compared with the results where experts’ distributions were combined with performance-based weights (IndepSLR-PW). We can note that IndepSLR-EW is much more spread out and quite different to IndepSLR-PW. This phenomenon is often encountered in expert judgment studies. Moreover, we can examine the summary statistics and plots (in Figure 6) of SLR distributions in the case of performance-based combinations with and without dependence. As expected, the means of both distributions are the same, but the distribution with dependence is more spread out than the one without dependence. Note the negative contribution to SLR for the 5th percentile value when dependence is included. This indicates a small probability that increased Accumulation can outweigh the mass loss due to Discharge and Run-off.

In Figure 7, bar plots of ExcProb of 0.5, 0.25, 0.05 and 0.01 for SLR are presented, with or without dependence. In the case of independence, the probability of exceeding 2000 mm of SLR is 0.01 which, in the dependent case, is significantly larger, at 0.048. Thus, the 0.01 ExcProb when dependence is taken into account corresponds to an SLR of about 2600 mm.

In Table 6, more detailed SLR results for the model with dependence are presented. The largest average contribution to the sea level rise is due to Discharge from the West Antarctic ice sheet, which also has the largest correlation with SLR. However, when influences due to Accumulation, Discharge and Run-off are combined, the largest average contribution is due to the Greenland ice sheet.

Let us examine in Figure 8 the scatter plots of E, G, W and SLR. We can see very complicated relationships between these variables, which are likely caused by mixing distributions of experts using performance-based weights. This phenomenon is most visible for pairs containing the distribution of Greenland. Two experts with the highest weight, expert 3 and 14, disagree significantly about the SLR values under this scenario. The scatter plot of distributions of G and SLR for both experts is shown in Figure 9. The average SLR according to expert 3 is 789 mm, which is almost twice the average assessed by expert 14, equal to 431 mm.

In Figure 9, the scatter plot of the distributions of Greenland’s contribution and the SLR for both experts 3 and 14, illustrating an example of inter-expert complexity mentioned above, are presented.

## 5. Discussion

To our knowledge, the recent ice sheet elicitation marks the first time expert tail dependence has been elicited and used in risk quantification. Including positive dependence and upper tail dependence increases the ‘fatness’ of the upper right tails of summed variables. This is clearly seen if we focus on the eight weighted experts for the case of the +5 °C scenario in 2100. These are experts for whom the hypothesis that their uncertainty quantifications are statistically accurate would not be rejected at the 1% significance level. With the exception of expert 8, all experts’ 95th percentiles increase as a result of the dependence elicitation. In many cases, the 5th percentiles also decrease. The medians remained the same or decreased (Figure 10). Note the diversity in the assessments of the ‘statistically accurate’ experts. Expert 8’s 95 percentile is just below 1 m, while that of expert 12’s is 2.1 m.

It is known that the experts’ performance is a persistent property and that performance-based combinations outperform equal weighting, both in- and out-of-sample [7]. Averaging experts (either with equal weights or performance weights) tends to blur this effect somewhat (Figure 11). The performance-based combination’s 95th percentile is 1.65 m. Experts 3 and 14 accounted for 52% of the weight in the performance-based combination (note the differences in their assessments).

As we become increasingly focused on the quality of uncertainty quantification, it behooves us to apply the best available techniques. The present study presents evidence that quantifying probabilistic dependence and tail dependence can be an important component of best practice (see also Appendix A).

## 6. Conclusions

The work presented here marks a further step in expert uncertainty quantification. From its inception, risk analysis has dealt with high-impact low-probability events involving inter alia nuclear safety, natural hazards (earthquakes and volcanoes) and public health. By their nature, these are always characterised by insufficient data, unvalidated models and an absence of ‘the’ relevant probability distributions. These features, sometimes characterised as ‘deep uncertainty’, are standard fare with which every risk analyst is required, and trained, to grapple. Expert uncertainty quantification has therefore been a cornerstone of risk analysis for over 50 years, punctuated by methodological advances including rigorous and traceable expert elicitation, the validation of expert subjective probabilities (in- and out-of-sample), performance-based weighted combinations, and uncertainty propagation with dependence (as chronicled in [14]).

Acknowledging the reliance on expert subjective probabilities disables the deflection that we ‘don’t know the probability distributions’—there is nothing to not know, there is only the quantification of subjective uncertainty (e.g., [15] Chs 2, 3 and 13). Now, as humanity faces existential risks from human-induced climate change, there is a temptation to revert to the 1960s’ unstructured approaches to uncertainty quantification, behind a fig leaf of ‘deep uncertainty’ (colloquially, BOGSAT: Bunch of Guys/Gals Sitting Around a Table). Without transparent elicitation, without the validation of subjective probabilities and without quantitative uncertainty propagation, this marks a degenerative methodological move which has significant potential to cause harm, which future generations will surely condemn. 

## Figures and Tables

**Figure 1 entropy-26-00949-f001:**
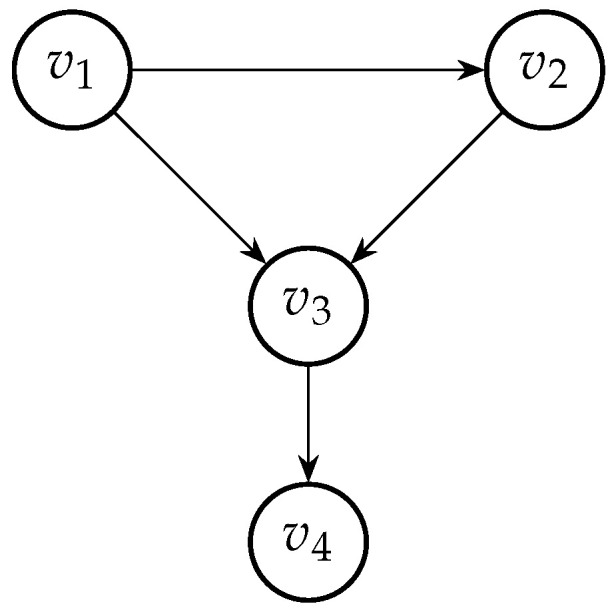
DAG with 4 nodes.

**Figure 2 entropy-26-00949-f002:**
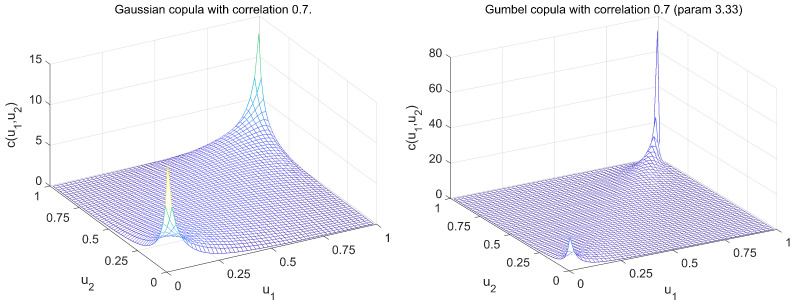
Gaussian and Gumbel copula densities both with correlation 0.7.

**Figure 3 entropy-26-00949-f003:**
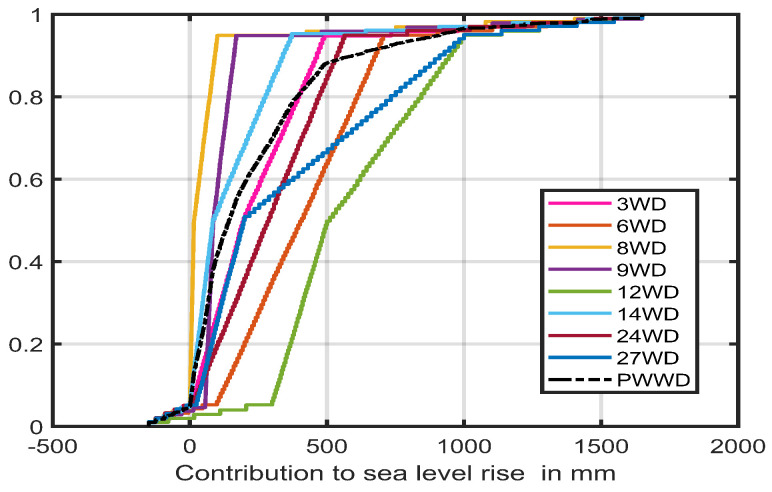
Cdf plots of marginal distributions of sea level contributions of West Antarctica due to Discharge assessed by experts 3, 6, 8, 9, 12, 14, 24 and 27, and the performance-based combination PWWD. The latter black dashed line is mainly hidden in the central region of the various expert Cdfs.

**Figure 4 entropy-26-00949-f004:**
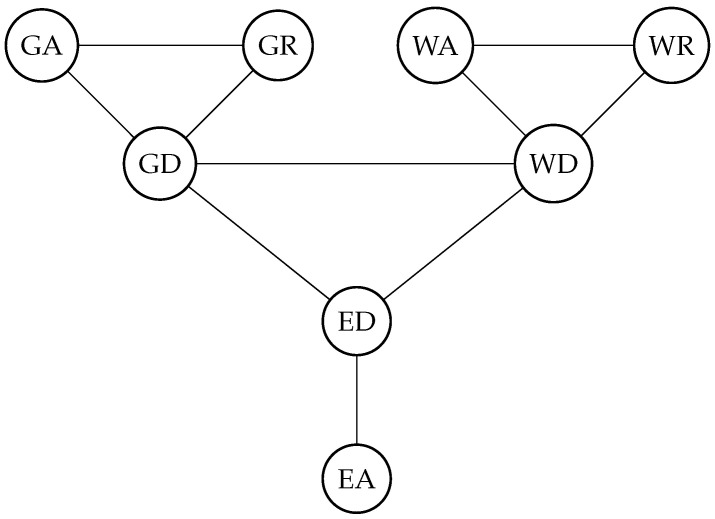
Graph of bivariate relationships assessed by experts.

**Figure 5 entropy-26-00949-f005:**
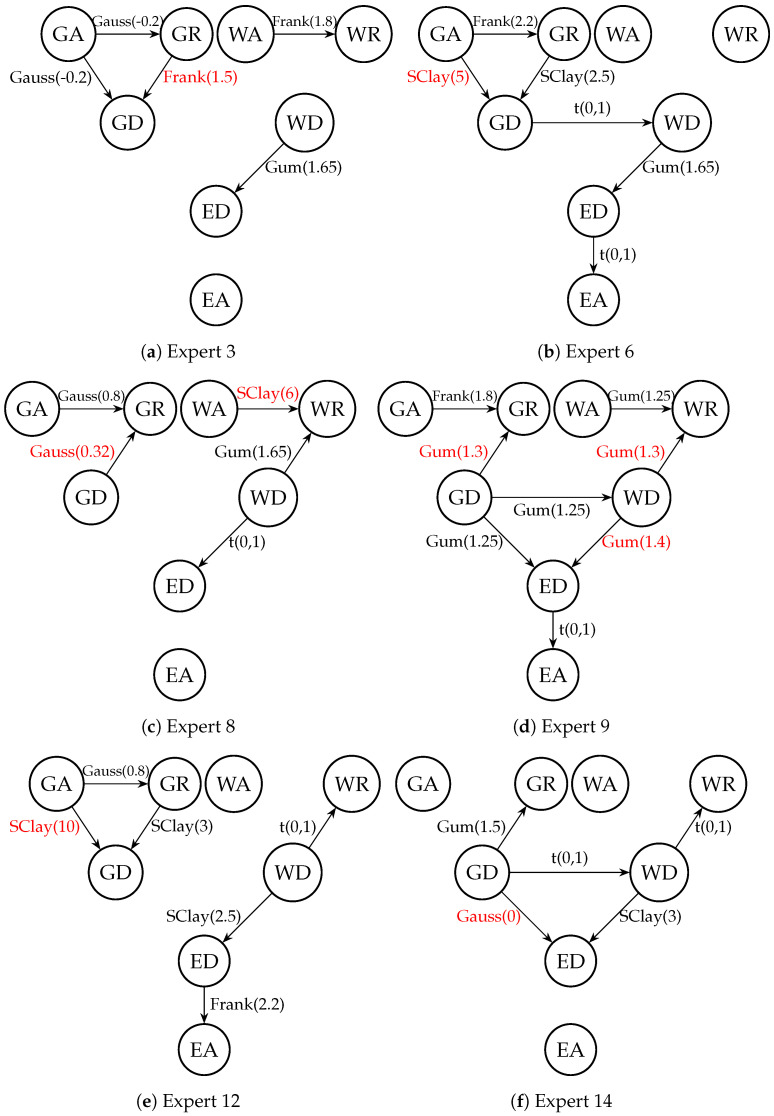
PCBN models of all experts with non-zero weights. Types of copulas and conditional copulas (in red) and values of copula parameters are assigned to arks of each graph.

**Figure 6 entropy-26-00949-f006:**
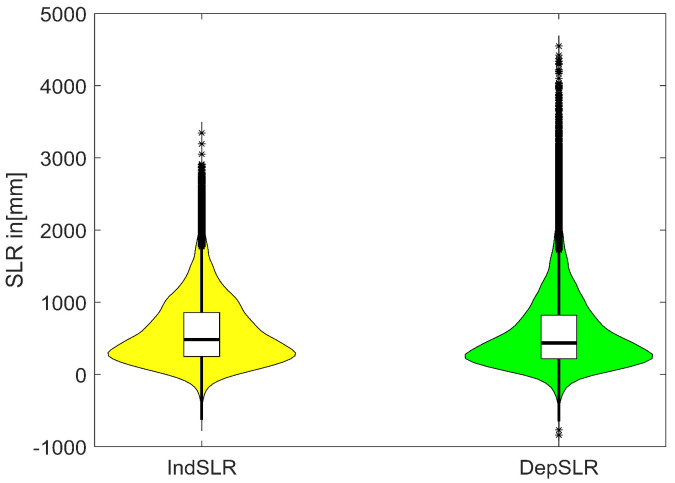
Violin plots of sea level rise distributions obtained with performance-based weights combination with dependence (DepSLR—green) and without (IndSLR—yellow).

**Figure 7 entropy-26-00949-f007:**
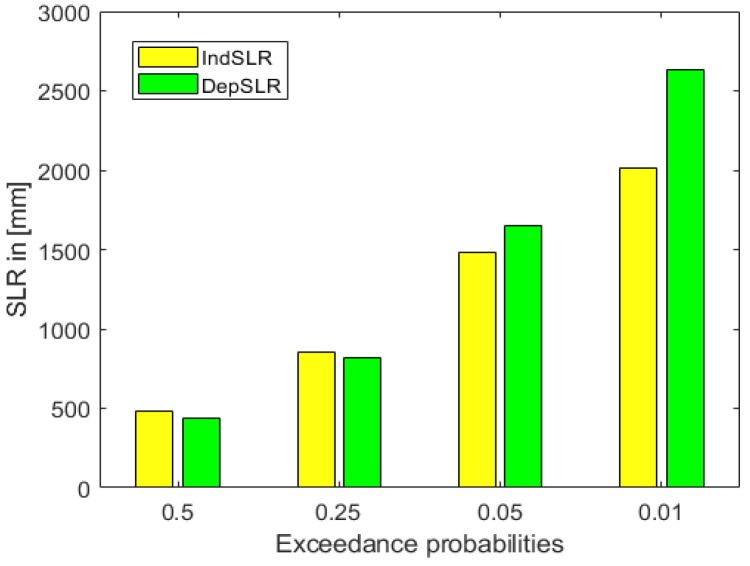
Bar plots of exceedance probabilities for SLR with (green) and without (yellow) dependence.

**Figure 8 entropy-26-00949-f008:**
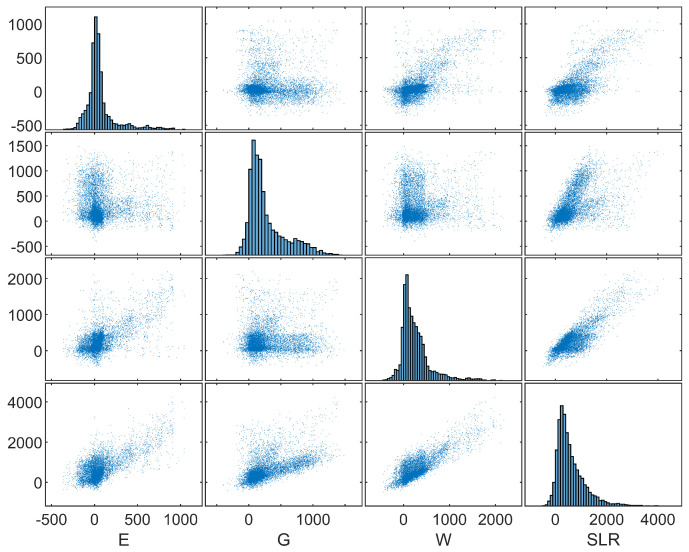
Scatter plot matrix of E, G, W and SLR.

**Figure 9 entropy-26-00949-f009:**
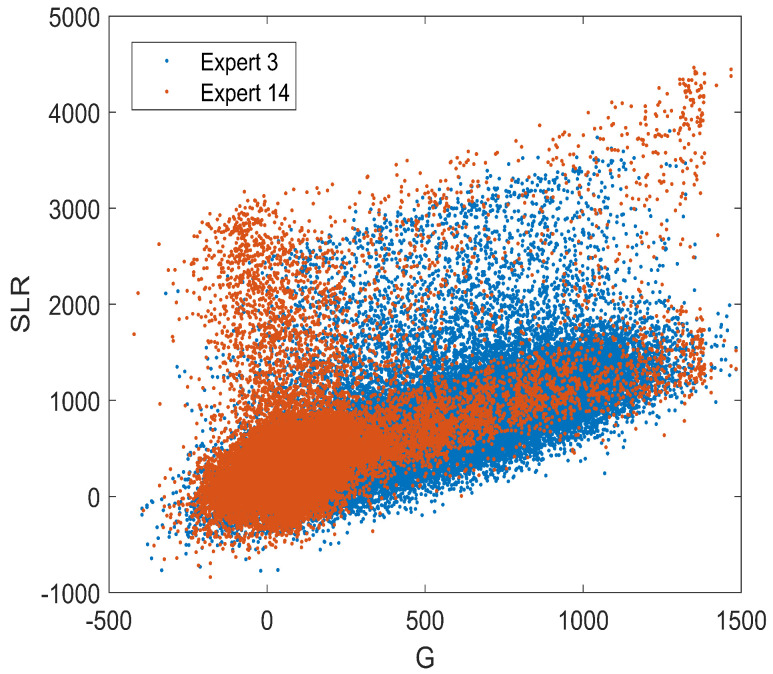
Scatter plot of G and SLR for experts 3 and 14.

**Figure 10 entropy-26-00949-f010:**
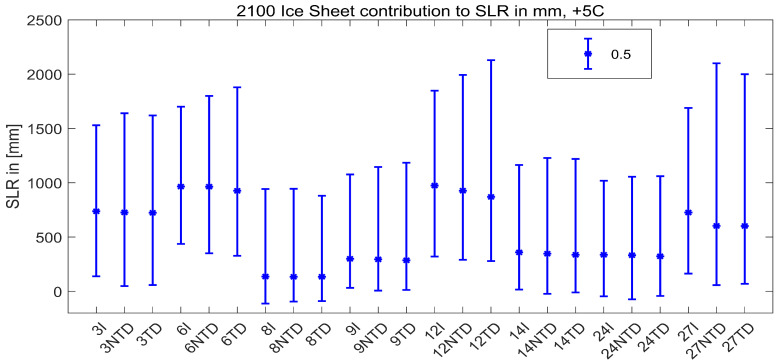
Effect of eliciting dependence and tail dependence of 8 weighted experts. Bars represent 5%, 50% and 95% of SLR distributions of the experts combined with equal (EW) and performance-based (PW) weights in the case of independence (I), Gaussian copulas corresponding to 50% ExcProb assessed by experts (NTD) and with copulas with tail dependence (TD), as discussed in the paper. Experts 3 and 14 with weights 0.28 and 0.3, respectively, are the most important experts.

**Figure 11 entropy-26-00949-f011:**
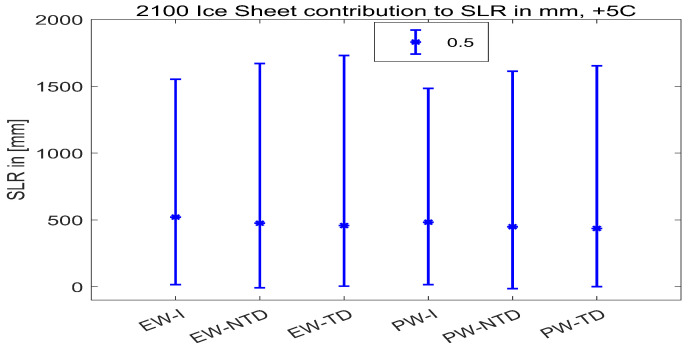
Effect of adding eliciting dependence and tail dependence on EW and PW DMs. Bars represent 5%, 50% and 95% of SLR distributions combined with equal (EW) and performance-based (PW) weights in case of independence (I), Gaussian copulas corresponding to 50% ExcProb assessed by experts (NTD) and with copulas with tail dependence (TD), as discussed in the paper.

**Table 1 entropy-26-00949-t001:** Weights of experts with non-zero weight.

Expert	Ex 3	Ex 6	Ex 8	Ex 9	Ex 12	Ex 14	Ex 24	Ex 27
Weights	0.28	0.03	0.11	0.07	0.09	0.30	0.04	0.08

**Table 2 entropy-26-00949-t002:** Colour coding dependencies.

Colour	Description	0.5 ExcProb	0.95 ExcProb
	Strong positive	0.8	0.8
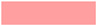	Positive	0.7	0.5
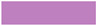	Weak positive	0.6	0.3
	Independent	0.5	0.05
	Weak negative	0.4	0.03
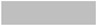	Negative	0.3	0.02
	Strong negative	0.2	0.01

**Table 3 entropy-26-00949-t003:** Copulas with parameters corresponding to colour coding.

0.5 ExcProb	0.95 ExcProb	Copula (prm)	Realised 0.5; 0.95	0.5 ExcProb	0.95 ExcProb	Copula (prm)	Realised 0.5; 0.95
		Indep	0.5; 0.05			Gum (1.25)	0.6; 0.3
		Frank (1.8)	0.61; 0.1			Gum (1.5)	0.67; 0.44
		Frank (2.2)	0.63; 0.11			Gum (1.65)	0.7; 0.5
		t (0,1)	0.5; 0.3			SClay (2)	0.75; 0.7
		Gauss (−0.2)	0.43; 0.02			SClay (2.5)	0.78; 0.76
		Gauss (0.6)	0.7; 0.31			SClay (3)	0.81; 0.8
		Gauss (0.8)	0.79; 0.5				

**Table 4 entropy-26-00949-t004:** Conditional copulas.

Expert	Conditional Copula	Conditional Copula (Type)	Unconditional Copula	50%; 95% Achieved	50%; 95% Target
3	GR,GD|GA	Frank (1.5)	Frank (1.5)	0.6; 0.1	0.6; 0.1
6	GD,GA|GR	SClay (5)	SClay (2)	0.79; 0.24	0.75; 0.7
8	GR,GD|GA	Gauss (0.32)	Frank (1.8)	0.61; 0.1	0.6; 0.14
	WR,WA|WD	SClay (6)	Gauss (0.8)	0.83; 0.5	0.8; 0.5
9	GR,GD|GA	Gum (1.3)	Gum (1.25)	0.6; 0.3	0.6; 0.31
	WR,WD|WA	Gum (1.3)	t (0,1)	0.6; 0.28	0.5; 0.3
	WD,ED|GD	Gum (1.4)	Gum (1.5)	0.66; 0.44	0.67; 0.44
12	GD,GA|GR	SClay (10)	Frank (1.8)	0.79; 0.22	0.61; 0.1
14	GD,ED|WD	Gauss (0)	t (0,1)	0.5; 0.27	0.5; 0.3
24	GR,GD|GA	Gauss (0)	t (0,1)	0.5; 0.22	0.5; 0.3
	WD,ED|GD	Gum (1.32)	Gum (1.25)	0.61; 0.29	0.6; 0.3
27	GR,GD|GA	SClay (10)	SClay (2)	0.73; 0.47	0.75; 0.7
	WR,WD|WA	SClay (10)	SClay (2)	0.73; 0.47	0.75; 0.7
	GD,EG|WD	Gum (1.3)	SClay (2)	0.75; 0.7	0.75; 0.7

**Table 5 entropy-26-00949-t005:** Comparison of SLR contributions from ice sheets [mm] (for +5 °C global surface air temperature rise by 2100) obtained from models with independence when experts’ assessments are combined with equal or performance-based weights and from a model which is performance-based with dependence.

	Mean	StDev	5th%ile	50th%ile	95th%ile
IndepSLR-EW	623	576	2	461	1728
IndepSLR-PW	589	467	15	483	1484
DepSLR-PW	587	553	0	436	1654

**Table 6 entropy-26-00949-t006:** Summary statistics based on the performance-based SLR model in [mm] with dependence. The last column contains the correlation of each ice sheet process component with SLR; correlations for SLR independence are presented in brackets. E, G and W denote distributions of SLR contributions from the East Antarctic, Greenland and West Antarctic, respectively, and A, D and R denote Accumulation, Discharge and Run-off contributions per ice sheet.

	Mean	StDev	5th%ile	50th%ile	95th%ile	Corr(.,DepSLR) (Corr(.,IndSLR))
EA	35	64	−33	27	164	0.1 (0.07)
ED	100	177	−20	42	510	0.75 (0.5)
GA	24	54	−35	17	137	0.1 (0.05)
GD	94	103	0	64	291	0.48 (0.34)
GR	201	233	2	109	738	0.5 (0.48)
WA	15	101	−173	22	119	0.03 (−0.03)
WD	247	310	1	142	910	0.83 (0.73)
WR	19	51	0	7	85	0.24 (0.16)
E	65	180	−135	26	457	0.7 (0.46)
G	271	288	−28	173	902	0.56 (0.46)
W	251	324	−70	169	900	0.82 (0.54)
DepSLR	587	553	0	436	1654	

## Data Availability

The raw data supporting the conclusions of this article can be made available by the authors on request.

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
