# Peer review of "Foundational Aspects for Incorporating Dependencies in Copula-Based Bayesian Networks Using Structured Expert Judgments, Exemplified by the Ice Sheet–Sea Level Rise Elicitation"

_entropy, 2024, doi:10.3390/e26110949_

Round 1
Reviewer 1 Report
Comments and Suggestions for Authors
This paper presents an interesting application for using a copula Bayesian network to model climate data. I find that the analysis to be sound and convincing. However, the presentation of the manuscript needs improvement.
The following issues should be addressed:
1. The main text should include a table summarizing all the climate terminologies discussed in the paper, e.g., SLR, SEJ, DM, etc.
2. The figures must be reformatted and redrawn at a higher resolution than 300 DPI. Currently, they are below the publication standard.
The following is a recommendation:
3. The authors could consider discussing/exploring the possibility of using machine learning-based copulas rather than parametric copulas. This would allow the dependency modelling to be more flexible and fit the data better. For example, see:
Ling et al., Deep archimedean copulas. NeurIPS 2020.
Comments on the Quality of English LanguageThe English of this manuscript is satisfactory.
Author Response
General comments and detailed comments in italic.

Reviewer 2 Report
Comments and Suggestions for Authors
In the file main.pdf

Author Response

(The authors gave the same response as above.)

Reviewer 3 Report
Comments and Suggestions for Authors
This manuscript discusses the use of structured expert judgment to quantify uncertainty, primarily focusing on how to leverage expert opinions to populate a Copula-based Bayesian network for predicting sea level rise due to climate change. Overall, the content is thorough and the methodology is innovative; however, several issues need to be addressed before formal acceptance:
1. The manuscript mentions that structured expert judgment is used to assess and quantify uncertainty, but this does not seem to be the main focus of the paper, leading to a lack of clarity. For example, how were the weights of the experts in Table 1 determined? How were the experts selected, and how many were chosen?
2. Similarly, the expression in Section 3.2 is somewhat confusing; it is unclear what the colored blocks in Table 3 represent.
3. In constructing the Bayesian network, were all potential dependency paths considered? How can the authors ensure that important dependencies are not overlooked? Were validation steps for the Bayesian network model included? How is the model's predictive performance assessed?
4. In Table 6, the correlation assumptions seem to have limited effects on the correlation coefficients in most cases. Please analyze the reasons for this.
5. Conclusion: I hope the authors can focus the conclusion section on the relevant findings of the paper. Specifically, what are the results of your probability analysis regarding sea level rise due to climate change using expert judgment to populate the Copula-based Bayesian network? Additionally, how can the algorithm be improved in the future, or what insights does this provide for further research?
Author Response
General comments and the detailed comments in italic.

Reviewer 4 Report
Comments and Suggestions for Authors
This manuscript incorporates the correlations and tail dependencies between pairs of processes (accumulation, discharge and run off) in the calculation or sea level rise distributions. This research may have great impact on addressing sea level rise challenges. Here are some comments:
(1) P7, Line 229 - Line 233 appears to require revision and correction of typos.
(2) P7 Line 233, please elaborate a little more on how different ordering influences the analysis.
(3) In section 2.4, more details on different types of copula (and how to choose) would be appreciated.
(4) As pointed in P14 L349, the experts with the two largest weights have the average SLRs that differ significantly. Using weighted average may still neglect some important information. Further investigation on the weights of the experts may be necessary.
Author Response

(The authors gave the same response as above.)

Round 2
Reviewer 1 Report
Comments and Suggestions for Authors
I think the authors have done a good job in the revision. I do not have further comments.
Reviewer 3 Report
Comments and Suggestions for Authors
The authors addressed all my questions.
Comments on the Quality of English LanguageThere are no major problems with the language.
Reviewer 4 Report
Comments and Suggestions for Authors
The revision is satisfactory.